# Adhesion and Surface Roughness of Apatite-Containing Carbomer and Improved Ionically Bioactive Resin Compared to Glass Ionomers

**DOI:** 10.3390/jfb14070367

**Published:** 2023-07-12

**Authors:** Handan Yıldırım Işık, Aylin Çilingir

**Affiliations:** 1Department of Restorative Dentistry, Faculty of Dentistry, Beykent University, 34500 İstanbul, Turkey; 2Department of Restorative Dentistry, Faculty of Dentistry, Trakya University, Balkan Campus, 22030 Edirne, Turkey

**Keywords:** glass–ionomer cement, shear bond strength, surface roughness, thermal cycling

## Abstract

The surface roughness of different glass–ionomer-based materials and their shear bond strength with a resin composite with and without thermal cycling were evaluated. Ketac Molar (KM, 3M ESPE, St. Paul, MN, USA), Glass Carbomer (GC, GCP Dental, Leiden, The Netherlands), Bioactive (BA, PULPDENT, Corporation, Watertown, MA, USA) and Fuji II LC (FJ, GC, Tokyo, Japan) were used to prepare the specimens and they were kept in distilled water at 37 °C for 24 h. The surface roughness of the specimens was measured with a profilometer (*n* = 6). A universal adhesive resin was applied on glass–ionomer materials and cylindrical universal composites were applied and polymerized, respectively (*n* = 16). The specimens were divided into two subgroups. The first subgroup was subjected to thermal cycling. Shear bond strength was investigated for both subgroups (*n* = 8). Stereomicroscopy and SEM examinations were performed. The roughest surface was obtained in the GC group (*p* < 0.05). The shear bond strength of the specimens without thermal cycling was higher than that of those with thermal cycling (*p* < 0.05). The lowest shear bond was measured in the GC group (*p* < 0.05). Although FJ, KM and BA have been observed to be suitable materials for clinical use, BA, in particular, is evidenced to become the best option among the materials we tested. GC cement’s long-term performance needs to be improved.

## 1. Introduction

With the developments in adhesive materials in current dentistry practices, more conservative approaches have been adopted and healthy dental tissues have been consequently preserved [1]. Restorative treatments applied in a single session are now preferred to ensure and protect the integrity of hard dental tissues [2]. Amalgam, composite and glass–ionomer cement (GIC) are restorative materials used in routine clinical applications [1]. 

Today, the use of aesthetic materials has become widespread. Tooth-colored materials such as resin composite and glass–ionomer cement are considered aesthetically preferable [3]. Composites are widely used in dentistry due to their aesthetic and high mechanical properties [4]. However, high failure rates have been reported due to polymerization shrinkage and the related discoloration of composites, loss of retention, water absorption, abrasion, microleakage and secondary caries [2,4]. The absence of fluoride release, when used in individuals with high caries risk and toxic properties of monomers, are among its undesirable features [2,5].

Glass–ionomer cement can be used as a restorative and base material, adhesive cement and fissure sealant [6,7,8]. It has advantages including ease of use, fluoride release, serving as a fluoride reservoir, being tooth-colored, having bonding capability and a close thermal expansion coefficient to dental tissues, biocompatibility, as well as disadvantages such as lower wear resistance, physical and mechanical properties compared with composites, and sensitivity to moisture during setting [9,10,11]. Therefore, studies have been carried out to improve their physical and mechanical properties and antibacterial activity [12,13]. More studies comparing these new materials are warranted. 

It is recommended to apply resin composite on top of the GIC in restorations (sandwich technique) to confer the beneficial properties of GIC, such as chemical bonding to dentin, fluoride release, biocompatibility and the properties of the composite such as aesthetics and better mechanical properties [14]. The sandwich technique is used for layered restorations wherein glass–ionomer cement is placed to replace dentin tissue and composite is used to replace enamel tissue [15]. This technique reduces the complexity of applying the composite involved in the incremental technique and can also prevent possible postoperative sensitivity [15]. Studies also show that the sandwich technique increases resistance to secondary caries by preventing microleakage [15]. 

Bonding of glass–ionomer with both dentin tissue and resin material is essential. The surface characteristics, such as roughness, can determine the quality and clinical behavior of the restorative material [12]. A smooth surface is required for the desired aesthetics, but it also prevents the formation of a coloring film layer and plaque retention. The surface smoothness also decreases the coefficient of friction, thus reducing the wear rate [16]. As a result, surface roughness is an important criterion affecting the restoration’s aesthetics, friction, wear, bacterial adhesion and optical properties.

The development of technology has led to the introduction of ionically bioactive materials similar to glass–ionomer and composites, which are useful in restorative and pediatric dentistry [17]. Since they can activate repair mechanisms and evoke a positive response from dental tissues, they are referred to as “bioactive” [17]. Currently, a new material, Activa™ BioActive-Restorative (Pulpdent Corp., Watertown, MA, USA) is being introduced to the market. According to the manufacturer, it combines the benefits of glass ionomers and the aesthetics and strength of the composites and mimics the properties of natural teeth. BA contains a bioactive ionic resin matrix with high release and recharge rate of fluoride (F^−^) calcium (Ca^2+^) and phosphate (PO_4_^3−^) ions. The rubberized resin is hard and durable and contains reactive glass ionomer fillers in addition to a high fluoride release rate [17]. Glass carbomer cements are monomer-free, carbomized and nano-glass restorative cements developed from a conventional glass–ionomer material containing hydroxyapatite and fluorapatite nanoparticles [18]. It has improved chemical and mechanical properties compared to conventional glass ionomers [18]. Glass carbomers contain nanoscale fluoro/hydroxyapatite particles that promote the remineralization of caries-damaged enamel and dentin [7].

Therefore, the objectives of this study were to compare the effect of thermal cycling on the bond strength of commonly used base materials, conventional glass ionomer cement (CGIC) (Ketac Molar Easymix) (KM), resin-modified glass–ionomer cement (RMGIC) (FUJİ II LC) (FJ), newly developed glass–ionomer materials, an improved ionically bioactive resin-modified glass-ionomer (BioACTIVE base/liner) (BA), glass–ionomer cement containing nanoparticle fluorapatite (FAp), hydroxyapatite (HAp) (Carbomer) (GC) with a universal resin composite (Essentia), and these materials’ roughnesses. The null hypothesis was that there would be no difference between RMGIC, CGIC and new types of glass ionomers in terms of roughness and bond strength to a universal composite with or without thermal cycles.

## 2. Materials and Methods

Ketac™ Molar Easymix (KM, 3M ESPE, St. Paul, MN, USA), Glass Carbomer (GC, GCP Dental, Leiden, The Netherlands), Bioactive Base/Liner (BA, PULPDENT, Corporation, Watertown, MA, USA) and Fuji II LC (FJ, GC, Tokyo, Japan) were used to prepare the specimens. The materials used in this study are shown in Table 1.

### 2.1. Preparation of Specimens 

To evaluate the shear bond strength, a silicone mold with an outer diameter of 35 mm and a length of 45 mm (Figure 1A) was used to prepare a total of 64 acrylic blocks with a diameter of 25 mm and a height of 20 mm. Acrylic blocks were removed from the mold after they had been cured (Figure 1B). 

Specimens were prepared by placing 5 mm × 5 mm glass–ionomer cement into the cylindrical spaces in the middle of the prepared acrylic blocks. Sixty-four specimens (*n* = 16) were prepared in four groups. The number of specimens was determined using power analysis. The materials placed in the acrylic blocks were covered with mylar strips (Hawe Stopstrip, Kerr Hawe, Bioggio, Switzerland) on the top and bottom surfaces of the mold and pressed between two glass slides. A smooth surface was obtained by applying light pressure on the glass slides and allowing excess material to flow. All materials were prepared following the manufacturer’s recommendations. The light-cured materials were polymerized over a glass plate with an LED light device (Valo Cordless, Ultradent, South Jordan, UT, USA) with a wavelength of 385–515 nm and a light intensity of 1000 mW/cm^2^. The transparent mylar strip was removed after the setting or curing of materials was completed.

A universal adhesive system (G-Premio bond, GC, Tokyo, Japan) was applied to the four glass–ionomer cements to evaluate the shear bond strength between the base and composite materials. The adhesive was polymerized for 10 s with a LED light device (Valo Cordless, Ultradent, South Jordan, UT, USA) following the manufacturer’s instructions. A plexiglass mold with a depth of 4 mm and a diameter of 2 mm was placed on the base materials and a universal composite (Essentia, GC, Tokyo, Japan) was placed in this mold in two increments after the adhesive application; it was polymerized for 20 s with an LED light source. All prepared specimens were kept in a humid environment at 37 °C for 24 h and randomly divided into two groups.

To evaluate the surface roughness of the materials, 24 samples in four groups (*n* = 6) were prepared with disc-shaped glass–ionomer cement. The specimens were prepared by packing uncured glass–ionomer cement into custom-made polytetrafluoroethylene molds with a diameter of 8 mm and thickness of 2 mm. Then, the materials were covered with mylar strips on the top and bottom surfaces of the mold and pressed between two glass slides. The specimens were prepared following the manufacturer’s instructions.

### 2.2. Thermal Cycle Test

Specimens were subjected to 5000 thermal cycles for 15 s between 5 and 55 °C with 20 s of dwell time (MOD Dental MTE-101, Esetron Mekatronik Ltd., Şti, Ostim, Ankara). 

### 2.3. Shear Bond Strength Test

The shear bond strength test was applied to all specimens using a universal testing machine with a head speed of 0.5 mm/min [19,20,21] (MOD Dental, Esetron Mekatronik Ltd., Şti, Ostim, Ankara, Turkey) (Figure 2A,B).

The flowchart in Figure 3 summarizes the used methodology for thermal cycling and the shear bond strength test (Figure 3). The shear bond strength (MPa) was calculated by dividing the load by the surface area. 

### 2.4. Stereomicroscopy and Scanning Electron Microscopy Analysis of Debonded Surfaces

After debonding, the fractured surfaces of all specimens were examined using stereomicroscopy (Leica L205 FA, Wetzlar, Germany) and one specimen from each group with scanning electron microscopy (SEM, Quanta FEG 250, FEI, Eindhoven, The Netherlands). The SEM micrographs were taken at ×50, ×70, ×80 and ×150 magnifications.

The failure modes evaluated were determined according to the following classification and the obtained fracture types were recorded for each specimen [19]:Type 1—cohesive failure inside the base material or composite;Type 2—adhesive failure at the interface of the base material and the composite;Type 3—mixed failure, combined failure (adhesive and cohesive).

### 2.5. Surface Roughness Test

Specimens prepared with four different glass–ionomer cements (*n* = 6, a total of 24 specimens) were kept in distilled water at 37 °C for 24 h. The surface roughness test was performed with a 2 µm contact-style mechanical profilometer device (Surtronic S128; Taylor Hobson, Leicester, UK) (Figure 2C,D). The cutoff distance was 0.8 mm and the measurement length was 1.5 mm in the profilometer device. Four-point measurements were taken by performing a 90° rotation on the measurement surface and the average surface roughness (Ra-µm) level was calculated for each specimen. The flowchart in Figure 4 summarizes the used methodology for the surface roughness test.

### 2.6. Statistical Analysis

Based on the power analysis (G*Power, ver. 3.1.9.7, Düsseldorf, Germany) when the effect size d (effect size): 0.628 and SD: 7.5 were taken for the bond strength, the number of samples determined for power: 0.80 and α: 0.05 was determined as a minimum of *n* = 7 for each group to evaluate the shear bond strength. The normality distribution of the data was evaluated with the Kolmogorov–Smirnov and Shapiro–Wilk tests and it was determined that the parameters were suitable for normal distribution. The one-way ANOVA test was used to evaluate shear bond strength and roughness according to materials. Tukey’s HSD and Tamhane’s T2 tests were used to determine the group that caused the difference. Student’s *t*-test was used to compare the parameters between the two groups. The chi-square test was used for in-group comparisons of the parameters. IBM SPSS Statistics 22.0 software (SPSS Inc., Chicago, IL, USA) was used for statistical analysis and a *p* value < 0.05 was considered statistically significant.

## 3. Results

### 3.1. Shear Bond Strength and Thermal Cycle Test

Table 2 represents the shear bond strength values before and after thermal cycling procedures. There was a statistically significant difference between the materials in terms of bond strength (*p*: 0.000; *p* < 0.05). Student’s t test revealed that only FJ’s shear bond strength was significantly higher before thermal cycling compared to other groups (*p*: 0.000; *p* < 0.05). There was no significant difference in the bond strength of the other groups (BA, KM, GC) with and without the thermal cycle (*p* > 0.05) (Table 2).

The one-way ANOVA test indicated a significant difference between the shear bond strength of the materials when the thermal cycling test was not applied (*p*: 0.000; *p* < 0.05). Tukey’s HSD test indicated that the GC’s shear bond strength was significantly lower than that of FJ, BA and KM (*p* < 0.05). KM’s shear bond strength was significantly lower than that of FJ and BA (*p* < 0.05). There was no significant difference between the bond strength of FJ and BA (*p* > 0.05) (Table 2).

There was a statistically significant difference between the shear bond strength of the materials when the thermal cycle test was applied (*p*: 0.000; *p* < 0.05) (Table 2). Tamhane’s T2 test showed that the bond strength of GC was significantly lower than FJ and BA (*p* < 0.05). The mean bond strength of KM was found to be significantly lower than that of BA (*p* < 0.05). There was no statistically significant difference between the bond strength of KM and both GC and FJ (*p* > 0.05) (Table 2).

### 3.2. Surface Roughness

One-way ANOVA showed that there was a statistically significant difference between the surface roughnesses of the materials (*p*:0.000; *p* < 0.05). The mean surface roughness of GC was significantly higher than FJ, BA and KM (*p* < 0.05) (Table 3). Tamhane’s T2 test showed that the mean surface roughness of BA was found to be significantly higher than that of KM (*p* < 0.05). There was no statistically significant difference between the mean roughness of FJ and BA (*p* > 0.05). There was no statistically significant difference between the mean roughness of FJ and KM (*p* > 0.05) (Table 3). The surface roughness of FJ was lower than GC’s. The surface roughness of FJ had no significant difference with both BA and KM (*p* > 0.05). BA had statistically significant higher surface roughness than KM and lower surface roughness than GC. The surface roughness of KM was lower than BA and GC (*p* < 0.05) and there was no significant difference with FJ (*p* > 0.05). The surface roughness of GC was higher than FJ, BA and KM (Table 3).

### 3.3. Stereomicroscopy and SEM Evaluations

Representative stereomicroscopy images of the different failure modes are presented in Figure 5.

SEM micrographs of the fractured surfaces of all groups at ×150 magnifications are presented in Figure 6 and Figure 7.

Fracture-type analysis without thermal cycling showed a significant difference between failure modes. An adhesive failure mode was more common in the GC group than in the BA group. Mixed failure was less common in the GC group than in the BA group. Cohesive failure was more common in the FJ group than in other groups (Table 4). 

Fracture-type analysis results after the thermal cycling test showed a statistically significant difference between failure modes. Adhesive failure was more common in the GC group compared to the BA group. Mixed and cohesive failure modes did not differ in the groups. There was no statistically significant difference in the comparison of thermal cycling and fracture types with and without the test (Table 4).

## 4. Discussion

The biocompatibility of resin materials containing toxic monomers such as HEMA, TEGDMA, UDMA and Bis-GMA is debated. These monomers are released from restorative materials and can diffuse into the pulp, gingiva, salivary and the human circulatory system. Cytotoxic conditions such as embryotoxicity caused by resin composites have been reported [22]. They can also cause various adverse biological effects, such as persistent inflammations, sensitivity and allergic reactions in patients [23]. Therefore, it is essential to use glass–ionomer cement as a base material, especially in deep cavities.

Using a base material reduces polymerization shrinkage stress and prevents the formation of gaps between the tooth and the restorative material, thus reducing microleakage [24]. However, a durable bond must be established between all materials to succeed in this application. In other words, the base material’s bond strength must be as strong with the resin composite as it is with the dental tissues [25]. Koç Vural et al. [24] showed that using a base material under different resin composites significantly reduced polymerization shrinkage.

Few studies have evaluated the bond strength between materials such as RMGIC, flowable composite and composite. To the best of our limited knowledge, there are no data regarding the shear bond strength of BioActive and composite, which has been introduced to the market in recent years and was one of the materials used in our study.

The thermal cycling test is commonly used to transfer intraoral temperature changes to in vitro conditions [19]. The temperature values to which thermal cycling will be applied should mimic the oral environment. The temperature values recommended by the American Dental Association (ADA, Acceptance Program Guidelines 2001) for bonding values and microleakage tests of adhesive materials to be used on dentin and enamel are between 5 and 55 °C. Although there is no standard for applying thermal cycling tests, 500–10,000 cycles are considered significant. It has been reported that 10,000 cycles are equivalent to a mean aging of 1 year [26,27]. There are also studies on the effects of thermal cycling on bond strength between base materials and resin composite and the surface characteristics of glass–ionomer cement on bonding. 

Bond strength tests are most commonly used to assess the adhesion properties of restorative materials. The shear bond strength test is an easy-to-apply and reliable test widely used to evaluate the bonding performance of restorative materials [28]. It is a preferred method, especially for glass–ionomer cement with lower bond strength, since it is difficult to perform other bonding tests [21]. A shear bond strength test was used in this study since shear forces were reported to be the stress to which restoration is mostly exposed [29].

Surface roughness is an important criterion when choosing a dental material [30]. It affects the long-term success of restorative materials by reducing their durability and surface quality [31,32]. Since glass–ionomer cement is structurally brittle and prone to porosity due to its powder–liquid formulation, this may result in a poor structure and failure in bonding [33]. Particle size or porosity distribution significantly affects cement durability and bonding [34,35]. Studies have reported that the smoothest surface is obtained when a mylar matrix is used in the finishing process [36,37]. Thus, a mylar matrix was used in the current study. Consequently, surface roughness and thermal cycling were also assessed in this study, in addition to the shear bond strength between the base materials and resin composite.

Manihani [38] found that the bond strength of CGIC to resin composite was lower than that of hybrid ionomers. As reported, RMGIC showed the highest bond strength with resin composite compared to CGIC and HVGIC (high-viscosity glass–ionomer cement) [38]. The increased bonding value can be explained by the chemical bonding of the unsaturated double bonds in the oxygen inhibition layer formed on the surface of the RMGIC with the adhesive and composite and the HEMA in the RMGIC, increasing wettability [39,40]. Similarly, the higher bond strength value of FJ compared to KM may be due to the HEMA in the resin structure in this study. In addition, inability to set the powder–liquid ratio correctly, as well as mixing and application times may affect the mechanical properties of the cement in manually mixed cement, such as the KM used in the present study.

In previous studies, thermal cycling decreased the shear bond strength of RMGIC to dentin tissue [40] and 5000 cycles of thermal cycling application significantly reduced the bond strength of all adhesive systems [41]. In other current research, bond strength is found to be lower after thermal cycling tests, similar to the results of this study. While the thermal cycle test decreased the bond strength of RMGIC with the resin composite, it had no significant effect on the bond strength of CGIC with the composite [41]. The abovementioned study’s findings are consistent with the results of the present study. It was observed that the shear bond strength of all glass–ionomer-containing materials used in the present study to a universal composite, Essentia, decreased after the thermal cycling test. However, this was statistically significant only in FJ. On the other hand, according to a meta-analysis, thermal cycling did not affect shear bond strength, which is inconsistent with the results of this study [42]. 

While the most frequently observed failure mode after thermal cycling was the adhesive failure in GC, adhesive failure was not observed in BA. Cohesive failure observed between the dental tissue and the dental material indicates that stress occurs within the material, preventing the correct evaluation of bond strength [43]. In past research, both failure modes between composite, RMGIC and CGIC were adhesive failures [39]. It was observed that the failure between FJ, KM and resin composite before thermal cycling was mainly due to adhesive reasons. In GC, adhesive-type failure was observed at a high rate before and after the thermal cycling test. This may be the chemical/mechanical interactions between the materials, and dentin bonding with the resin composite is mostly micromechanical. This possibility is strengthened by the fact that glass carbomer had the highest number of micropores in our SEM analysis and because the roughest surface was observed in GC. While Panahandeh et al. [44] found similar results as those of present study in terms of RMGIC’s higher bond strength values, they reported different results in terms of CGIC with a rougher surface, which had a higher bond strength than CGIC with a surface of lower roughness. According to the results of this study, the GC with the roughest surface and highest porosity had the lowest shear bond strength values. No statistically significant correlation was found between surface roughness and bond strength. Thus, the null hypothesis was rejected. 

This study had some limitations. Due to the preparation of the specimens used in the mold, the materials and bonding surfaces came into direct contact with water during thermal cycling. Although it is reinforced, direct contact of Ketac Molar with moisture can affect its water absorption and dissolution. Preparing specimens with human teeth instead of only materials can increase the reliability of the results.

This in vitro study was unable to mimic biological changes, such as chewing forces that would impair the durability of the restoration and chemical attack by acids and enzymes. Therefore, future studies are needed on physical, mechanical, biological and clinical properties, as well as evaluations of cell cytotoxicity of BioActive.

## 5. Conclusions

The findings of this study indicate that the shear bond strength of GC was significantly lower than that of BA, FJ and KM before and after the thermal cycling procedure. GC’s roughness was significantly higher than FJ and KM. Based on the findings, BioActive, which offers strengthened mechanical properties unlike conventional glass–ionomer structures, may be considered the best option for taking advantage of glass–ionomer cement and eliminating the adverse effects of resin materials; on the other hand, GC cement’s long-term performance needs to be improved.

## Figures and Tables

**Figure 1 jfb-14-00367-f001:**
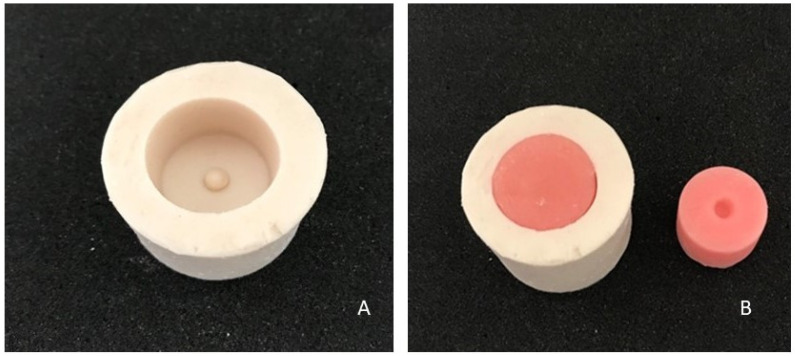
The molds used in the preparation of the specimens. (**A**): Silicone mold. (**B**): Acrylic blank block.

**Figure 2 jfb-14-00367-f002:**
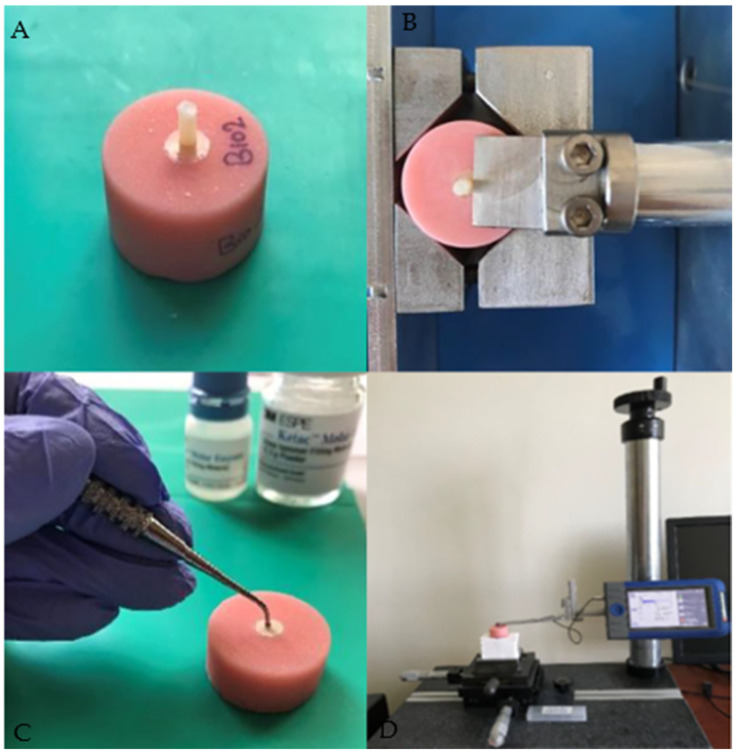
(**A**) Prepared specimen for shear bond strength test. (**B**) Shear bond strength test. (**C**) Prepared specimen for the surface roughness evaluation. (**D**) Surface roughness measurement.

**Figure 3 jfb-14-00367-f003:**
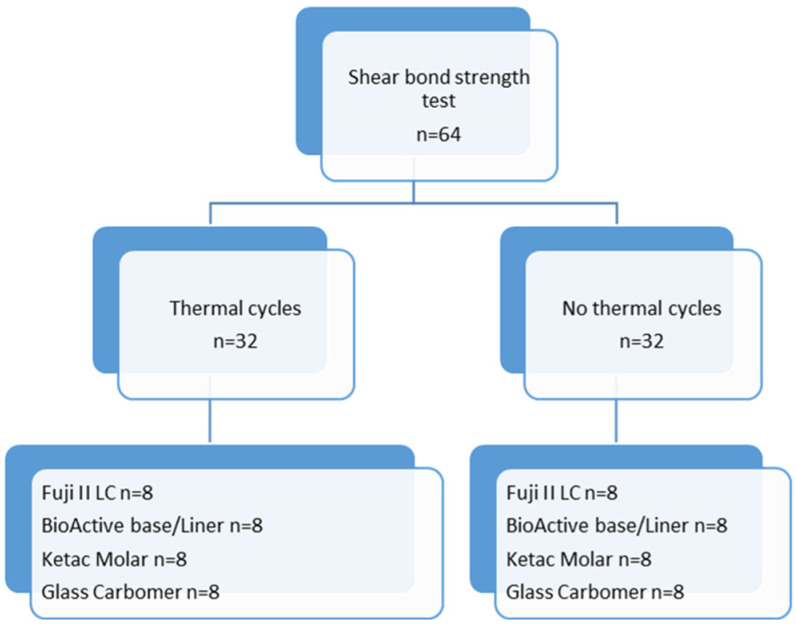
Flowchart summarizing thermal cycling and shear bond strength test.

**Figure 4 jfb-14-00367-f004:**
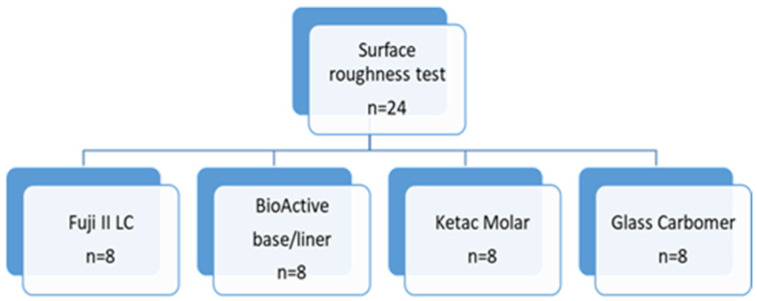
Flowchart summarizing the surface roughness test.

**Figure 5 jfb-14-00367-f005:**
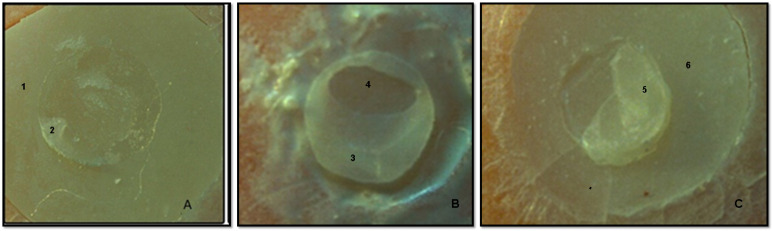
Stereomicroscopy images of the fractured surfaces of the specimens used in this study. (**A**) Adhesive-type failure, (**B**) cohesive-type failure and (**C**) mixed-type failure. Numbers 1, 4 and 6 represent the different types of glass–ionomer base materials. Numbers 2, 3 and 5 represent the remnants of the universal composite Essentia.

**Figure 6 jfb-14-00367-f006:**
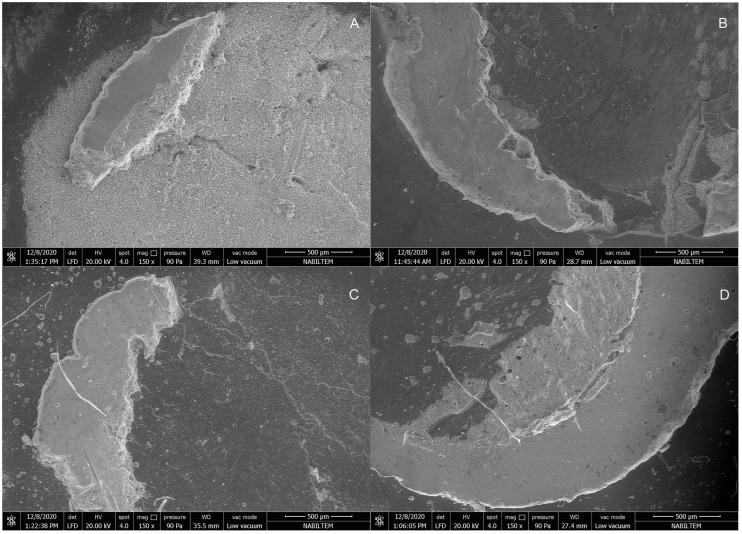
SEM micrographs of the fractured surfaces of all specimens at ×150 magnification. (**A**) Ketac Molar, (**B**) Fuji II LC, (**C**) BioActive, (**D**) Glass Carbomer.

**Figure 7 jfb-14-00367-f007:**
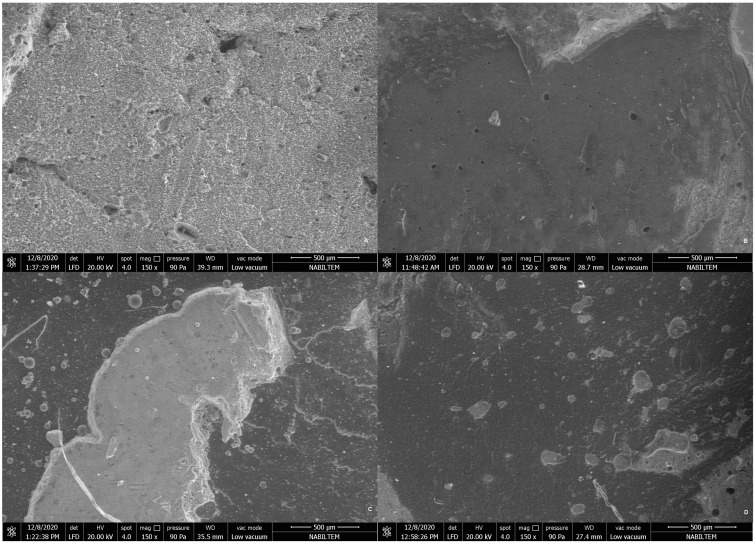
SEM micrographs of the fractured surfaces of specimens. (**A**) Ketac Molar, (**B**) Fuji II LC, (**C**) BioActive, (**D**) Glass Carbomer.

**Table 1 jfb-14-00367-t001:** Materials used in this study.

Materials	Manufacturer	Composition	Lot
Conventional glass–ionomer cement, Ketac Molar Easymix	3M ESPE, St Paul, MN, USA	Powder: aluminum–calcium–lanthanum fluorosilicate glass, acrylic acid, maleic acid; Liquid: poly(alkenoic acid) tartaric acid, water.	6702046
Resin-modified glass–ionomer cement, Fuji II LC	GC, Tokyo, Japan	Powder: fluoroaluminosilicate glass, HEMA, urethane dimethacrylate, water, photoinitiator (camphoroquinone); Liquid: poly(acrylic acid).	191209A
Improved resin-modified glass–ionomer cement, BioACTIVE Base/Liner	PULPDENT, Corporation, Watertown, MA, USA	Urethane dimethacrylate, bis 2-methacryloyloxy ethyl phosphate, barium glass, poly(acrylic acid) maleic acid, copolymer, sodium fluoride, coloring agent, photoinitiator.	191009
Glass carbomer cement, GCP Glass Fill	GCP Dental, Leiden, Holland	Powder: fluoroaluminosilicate glass, apatite; Liquid: poly(acids).	71808616
Universal composite, Essentia	GC, Tokyo, Japan	UDMA, Bis-MEPP, Bis-EMA, Bis-GMA, TEGDMA, barium glass, silicon dioxide, coloring agent, photoinitiator.	191003A
Universal adhesive resin, G-Premio Bond	GC, Tokyo, Japan	MDP, 4-MET, MEPS, dimethacrylate monomer, acetone, water, silicon dioxide, photoinitiator.	1910244
Sealant, GCP gloss	GCP Dental, Leiden, Holland	Modified polysiloxane.	1607101

HEMA: 2-hydroxyethyl methacrylate; UDMA: urethane dimethacrylate; Bis-GMA: bisphenol A glycidyl methacrylate; TEGDMA: triethylene glycol dimethacrylate; Bis-EMA: bisphenol A dimethacrylate; Bis-MEPP: bisphenol A ethoxylate dimethacrylate, MDP: methacrylol oxidecyl dihydrogen phosphate, 4-MET: 4-[2-(methacryloyloxy) ethoxycarbonyl]phthalic acid, MEPS: methacryloyloxyalkyl thiophosphate methylmethacrylate.

**Table 2 jfb-14-00367-t002:** (**a**) Evaluation of shear bond strength according to thermal cycling test application and to materials. (**b**) Post-hoc assessment for the comparison of bond strength of materials with and without thermal cycling.

(**a**)			
	**Before TC** **Mean ± SD**	**After TC** **Mean ± SD**	** *p* **
Fuji II LC	35.68 ± 6.74	21.62 ± 4.99	0.000 *
BioActive	33.17 ± 5.02	30.93 ± 7.26	0.484
Ketac Molar	19.84 ± 7.66	14.73 ± 4.81	0.133
Glass Carbomer	11.71 ± 2.96	8.87 ± 2.87	0.072
*p*	0.000 *	0.000 *	
(**b**)			
		**^1^ With TC**	**^2^ Without TC**
Fuji II LC	BioActive	0.829	0.064
	Ketac Molar	0.000 *	0.080
	Glass Carbomer	0.000 *	0.000 *
BioActive	Fuji II LC	0.829	0.064
	Ketac Molar	0.001 *	0.001 *
	Glass Carbomer	0.000 *	0.000 *
Ketac Molar	Fuji II LC	0.000 *	0.080
	BioActive	0.001 *	0.001 *
	Glass Carbomer	0.046	0.073
Glass Carbomer	Fuji II LC	0.000 *	0.000 *
	BioActive	0.000 *	0.000 *
	Ketac Molar	0.046 *	0.073

(**a**) Student’s *t*-test. One-way ANOVA test. * *p* < 0.05. TC: thermal cycling test. Shear bond strength was evaluated in terms of thermal cycling in rows and material in columns. (**b**) ^1^ Tukey’s HSD test. ^2^ Tamhane’s T2 Test. * *p* < 0.05.

**Table 3 jfb-14-00367-t003:** (**a**) Evaluation of surface roughness according to materials. (**b**) Post-hoc assessment for the comparison of surface roughness of materials.

(**a**)		
		**Roughness** **Mean ± SD**
Fuji II LC		0.81 ± 0.26
BioActive		1.06 ± 0.38
Ketac Molar		0.79 ± 0.21
Glass Carbomer		2.03 ± 0.41
*p*		0.000 *
(**b**)		
		**Roughness**
Fuji II LC	BioActive	0.061
	Ketac Molar	1.000
	Glass Carbomer	0.000 *
BioActive	Fuji II LC	0.061
	Ketac Molar	0.023 *
	Glass Carbomer	0.000 *
Ketac Molar	Fuji II LC	1.000
	BioActive	0.023 *
	Glass Carbomer	0.000 *
Glass Carbomer	Fuji II LC	0.000 *
	BioActive	0.000 *
	Ketac Molar	0.000 *

(**a**) One-way ANOVA test. * *p* < 0.05. (**b**) Tamhane’s T2 Test. * *p* < 0.05.

**Table 4 jfb-14-00367-t004:** Distribution of fracture types in study groups with and without the thermal cycle test.

Group	Fracture Type	Test		Test Statistics
		With TC	Without TC	
Ketac Molar	Adhesive	6 (%75) ^a,b^	3 (%37.5) ^a,b^	χ^2^ = 1.000; *p* = 0.317; V = 0.333
	Mixed	1 (%12.5) ^a,b^	2 (%25) ^a^	χ^2^ = 0.333; *p* = 0.564; V = 0.191
	Cohesive	1 (%12.5) ^a^	3 (%37.5) ^a^	χ^2^ = 1.000; *p* = 0.317; V = 0.333
Fuji II LC	Adhesive	4 (%50) ^a,b^	2 (%25) ^a,b^	χ^2^ = 0.667; *p* = 0.414; V = 0.272
	Mixed	2 (%25) ^a,b^	2 (%25) ^a^	χ^2^ = 0.001; *p* = 0.999; V = 0.001
	Cohesive	2 (%25) ^a^	4 (%50) ^a^	χ^2^ = 0.667; *p* = 0.414; V = 0.272
BioActive	Adhesive	1 (%12.5) ^a^	0 (%0) ^a^	χ^2^ = 0.333; *p* = 0.564; V = 0.211
	Mixed	6 (%75.0) ^a^	4 (%50) ^a^	χ^2^ = 0.400; *p* = 0.527; V = 0.192
	Cohesive	1 (%12.5) ^a^	4 (%50) ^a^	χ^2^ = 1.800; *p* = 0.180; V = 0.407
Glass Carbomer	Adhesive	7 (%87.5) ^b^	6 (%75) ^b^	χ^2^ = 0.077; *p* = 0.782; V = 0.092
	Mixed	0 (%0) ^a^	1 (%12.5) ^a^	χ^2^ = 0.333; *p* = 0.564; V = 0.192
	Cohesive	1 (%12.5) ^a^	1 (%12.5) ^a^	χ^2^ = 0.001; *p* = 0.999; V = 0.001
Test statistics		χ^2^ = 15.906; *p* = 0.014; V = 0.476	χ^2^ = 14.409; *p* = 0.025; V = 0.409	

TC: thermal cycling; χ^2^: chi-square test; V: Cremer effect size; summary statistics are provided as number (percentage) values; ^a^ > ^b^: different letters or letter combinations on the same line represent the statistically significant difference (*p* < 0.05).

## Data Availability

The data presented in this study are available upon request from the corresponding author.

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
