# Peer review of "Adhesion and Surface Roughness of Apatite-Containing Carbomer and Improved Ionically Bioactive Resin Compared to Glass Ionomers"

_jfb, 2023, doi:10.3390/jfb14070367_

Round 1
Reviewer 1 Report
Dear authors,
please find my comments regarding the corrections of your manuscript:
-"bioactive resin and 2 fluoropatite and hydroxyapatite containing glass ionomers" rearrange the title, sounds incomprehensible
-"Assistant Professor": the property of the authors is not required, please replace with affiliation (lab/section, institution, address ...)
-"glass ionomer, thermal cycling": replace with "glass-ionomer", "thermal-cycling where needed in text. Use a dash to unite two nouns in raw
-"Bioactive, and Fuji II LC": never use a comma (,) before the words "and" and "or" when for simple parathesis of similar things. Check and correct throughout the text where needed.
-"hours, seconds, minutes": use SI symbols for units, check and replace in text with "h, s, min"
-"composite resins": the term is incorrect, replace with "composites" or "resin composites"
-abstract: rearrange a bit the sentence regarding the statistical analysis.
-"ideal base material.": what do you mean ideal? Perhaps the best option among the products you tested.
-"adhesive techniques": do you mean "adhesive-manufacturing techniques" or "adhesive-applying techniques"
-check the uniformity of citing in text and whether you have followed the journal's instructions.
-"fluorapatite (FAp) and hydroxyapatite (HAp) (Carbomer) (GC)": be more detailed on the ingredients. What does Carbomer stand for?
-"Polyalkenoic acid,": correct "Poly(alkenoic acid)", "Poly(acrylic acid)", poly(acrylic-co-maleic acid) etc. The fact that the companies write the ingredients as they think, does not mean that the scientific publications carry the mistakes on!
-Table 1: pH=1.5, where does that refer to? To what solution?
-check that subparagraph titles are according to format (2.1 Preparation of specimens etc). The figure legends too.
-"acrylic resins": incorrect, use "composite samples" where needed
-"37ºC", replace with "37 oC". Check and correct where needed in text.
-"The data for each sample was recorded in Newtons" remove the phrase, the force in any dynamometer where any mechanical test is operated is measured in N, yet depending on the test type the respective values are calcualted
-Tables: too much blank space in tables, reduce the size of the columns to the content
-the decimal digits in English language where the manuscript is written are pointed with dot (.). So correct all numbers
-explain/comment on the difference between adhesive and cohesive (table 4)
-"resin monomers": incorrect, just "monomers"
-Check if references are written according to journal's instructions and correct
Please take more care of your manuscript to show the quality of the experimental work. Do not emphasize on the statistics more than the materials' behaviour.
-"It has such advantages": do you mean "It has many advantages such as.."?
-"as disadvantages such as": better "as disadvantages, such as"
-"post-operative dentistry"
-"Stereomicroscope": replace with "Stereomicroscopy and ...microscopy..."
Author Response
Dear reviewer
revisions are atteched as a file
Best regards.

Reviewer 2 Report
The presented article deals with issues that have been repeatedly described and presented in other studies. For this reason, I rate the impact on knowledge of the subject as low.
The study itself was properly planned. A roughness test and a fatigue test were carried out, which should be considered as performed correctly.
However, I have big reservations about the microscopic research department. Using an SEM microscope that is capable of 100,000x magnification for 50x images is an absolute misunderstanding. This is not a device for that. Both streoscope and SEM images show nothing. We don't see any breakthroughs, we don't see the place of preparation, we don't see the place where the layers are joined. Additionally, the preparations are unprocessed and basically worthless.
In addition, the bibliography is outdated - a lot of items are from the period 2000-2002. This requires correction.
Therefore, the results may be distorted - this requires verification, and any summaries regarding the assessment of the surface structure based on the presented photos are undocumented. This requires verification by the Authors and improvement.
Author Response
Dear reviewer,
revisons are attached as a file
Best regards

Reviewer 3 Report
the paper needs revision for gramer and style, many typos
Abstract, add the detail of the new GIC used, sample size, add brand names, company name, city, country, and variables measured (Thermal cycle test, Roughness, Shear bond strength test, The failure modes), add the details of the statistical differences
Introduction/Discussion, very deficient, read and cite papers that reported on modifications/approaches to improve the properties of the GIC types of cement (J Prosthet Dent. 2017;118(1):102-107.;Dent Mater J. 2016;35(5):817-821.:Int J Periodontics Restorative Dent. 2016;36(3):425-30.)
Objectives and null hypotheses need revision 'in terms of their roughness and bonding capability and the effect of thermal cycles on the bond strength to a universal resin composite (Essentia) in this study. The hypothesis tested was that the new type of glass ionomers show less roughness and better bonding to resin composite.'
M&M, add the detail of sample preparation, detail of specimen dimensions, how you prepared them, add photos, sample size calculation
needs improvement
Author Response
dear reviewer,
revisions are attached as a file
best regards

Reviewer 4 Report
In this study, the surface roughness and bond strength of a new type GIC was investigated. Some issues need to be addressed.
Introduction
1. “Resin composite” and “composite resin” are used in the text. Actually, they are not equivalent. Please check you use both of them or “resin composite” only. If you use both of them, you may need to define them as some readers may confuse them. You can read this Editorial for your reference:
a. Watts DC. Resin composite or composite resin? Dental Materials 2020; 36: 1115.
2. It is suggested to discuss briefly the improved bioactive resin-modified glass ionomer (BMGIC). What are the improvements of the BMBIC compared to conventional GIC?
Materials and Methods
1. It is suggested to list each test group for the shear bond strength test, e.g.,
Group 1: Fuji II LC, Group 2: BioActive Base/Liner, etc.
2. What is the sample size of each group for shear bond strength test?
Discussion
1. From the results, there is no direct relationship between the surface roughness and the bond strength. This may be different from other studies.
2. Line 306, “In addition, the inability to set the powder-liquid ratio correctly, mixing, and application times may affect the mechanical properties of the cement in manually mixed cement such as KM in the present study.”. Why were you unable to set the powder-liquid ratio correctly?
3. Line 341, “….. direct contact of Ketac Molar with moisture can affect its water absorption and dissolution.”. The samples were subjected to thermal cycling to simulate artificial aging. This evaluated the bond degradation over time under aging as the water diffused into the bonding interface and caused hydrolytic degradation of the bonding.
4. Line 345, you can measure the surface micro-hardness and wear resistance. These parameters are related to chewing and biting.
5. The last paragraph, “Authors should discuss the results ……… be highlighted.”. what did you mean? You have discussed the results of this study.
Conclusions
1. Line 359, “ ….. eliminate the adverse effects of resin materials, …”. This may need to evaluate the cell cytotoxicity of the BioActive Base/Liner in future study.
1. English editing is recommended.
Author Response
Dear reviewers,
revisions are attached as afile
Best regards

Round 2
Reviewer 1 Report
Some comments:
-add full brands in Abstract
-"biocompatibility, and the properties", " adhesion, and optical properties": never use a comma (,) before the words "and" or "or" when for simple parathesis of similar things. Check and correct throughout the text
-The "sandwich" technique, methaphorical
-replace with "ionically bioactive resin", better
-Phosphate-releasing is more known, explain more the phosphate-charging process!
-explain what does "carbomer" mean or stands for
-"to megapascals" delete
-delete blank spaces among paragraphs and spaces
-with dash please, Bis-GMA
-check spacelines in references
good level
Author Response
"Please see the attachment"
Best regards

Reviewer 2 Report
the changes introduced in the study go in the right direction.
Unfortunately, my doubts were not dispelled by the authors. The article lacks a chapter in which the methodology of elaborating the results will be shown. Please create a subchapter in which it will be shown in detail, using an example, how the results in the table were created.
Still SEM is used incorrectly. All photos with his participation should be removed from the study.
Author Response
"Please see the attachment"
Best Regards
